# Nutrient Intake, Performance, Carcass Characteristics, Meat Quality, and Cost Analysis of Sheep Submitted to Intermittent Supplementation on Masai Grass Pastures

**DOI:** 10.3390/ani13071267

**Published:** 2023-04-06

**Authors:** Stela Antas Urbano, Jessica Caroline Nascimento Rodrigues, Pedro Henrique Cavalcante Ribeiro, Yasmin dos Santos Silva, Rodrigo Ferreira da Silva, João Virgínio Emerenciano Neto, Adriano Henrique do Nascimento Rangel, Juliana Paula Felipe de Oliveira, Henrique Rocha de Medeiros

**Affiliations:** 1Academic Unit Specialized in Agrarian Sciences, Federal University of Rio Grande do Norte, Macaíba 59280-000, RN, Brazil; 2Department of Animal Science, Federal University of Sergipe, Nossa Senhora da Glória 49680-000, SE, Brazil

**Keywords:** sheep meat, intake behavior, shear force, tropical grass, economic viability

## Abstract

**Simple Summary:**

Lowering the slaughter age of sheep is an indispensable measure to improve the quality of meat reaching the market and to increase the *per capita* consumption of this type of protein. Sheep meat production systems on pasture are less costly alternatives and associating them with supplementation strategies to reduce costs may be the path to consolidate the productive chain of sheep meat. We assessed intermittent supplementation (every day or every other day) in sheep managed with Masai grass and found no effect of said grass on nutrient intake, performance, intake behavior, carcass characteristics, or meat quality. However, it was seen that the strategy reduces the cost of producing 1 kg of meat by 15%, which encourages us to recommend the application of that nutritional strategy.

**Abstract:**

This study aimed to assess nutrient intake, performance, intake behavior, carcass characteristics, and meat quality of sheep managed on Masai grass pastures along with intermittent concentrate supplementation, in addition to the economic impacts of applying that strategy. A sample of 24 Santa Inês sheep (12 males and 12 females) was kept for 80 days on *Panicum maximum* cv. Massai pastures and was supplemented daily or every other day. The voluntary intake of dry matter and nutrients was estimated. Mean daily weight gain and total weight gain were calculated and intake behavior parameters were assessed. The males were slaughtered and the weight and yield of the carcasses and meat cuts were assessed. The meat was analyzed for cooking losses, shear force, and color. The gross margin of the system was estimated from the difference between total income and operational cost. No effect was seen of the interaction between supplementation frequency and sex for any parameter assessed (*p* > 0.05). Intermittent supplementation of Santa Inês sheep managed with Masai grass is recommended since the strategy does not impact nutrient intake, performance, intake behavior, carcass characteristics, or meat quality, but lowers the costs of producing 1 kg of meat by 15%.

## 1. Introduction

Sheep meat has great potential for large-scale commercialization as it is a very appreciated type of protein in haute cuisine restaurants. Nonetheless, the activity still faces bottlenecks in product standardization, which hinders competitiveness with other agribusiness segments. According to Urbano et al. [1], carcasses lacking qualitative standardization from animals slaughtered at an advanced age routinely enter the market, which results in undesirable organoleptic characteristics of the meat and harms the acceptability of the final product. Thus, the need arises to use nutritional management strategies that enable rapid finishing of lambs, leading to lower slaughter age and to the production of higher quality meat.

Pasture-based sheep meat production systems have become common in semi-arid regions due to the economics of providing nutrients to domestic ruminants via forage [2]. However, it is possible that anatomic characteristics of the cell wall of tropical grasses restrict access by ruminal microorganisms to cell contents [3], which compromises nutrient intake and animal performance. In this context, it is speculated that increasing microbial protein synthesis by providing substrates fermentable in the rumen may contribute to increasing the rate of fiber breakdown, thus shortening the time of forage retention in the reticulum-rumen compartment and increasing animal performance [4].

Supplementing animals kept on pasture with concentrate feed tends to maximize the use of the potentially digestible portion of dry matter since it provides energy and nitrogen to the ruminal environment and favors microbial growth, in turn enhancing digestibility and, consequently, dry matter intake [5]. However, the use of supplementation increases production costs, which entails the need for developing nutritional management alternatives that focus on economic viability. Hence, Moraes et al. [6], based on the physiological particularity of ruminant animals to maintain nitrogen supply in the ruminal environment via endogenous recycling at sufficient levels to maintain the dynamics of the microbial population, endorsed the positive results of supplying supplements with alternate frequencies found by Simioni et al. [7] and Canesin et al. [8].

Therefore, the intermittent supply of concentrate feed to animals reared on pasture may be able to lower costs without harming performance given the reduction with labor expenses. In face of that, this study aimed to assess nutrient intake, performance, intake behavior, and carcass characteristics, and meat quality of sheep managed on Masai grass pastures along with intermittent concentrate supplementation, in addition to the economic impacts of applying that nutritional strategy.

## 2. Material and Methods

The present research was submitted to evaluation by the Ethics Commission on Animal Use of UFRN and was approved under certificate 068/2018. This study was conducted in accordance with the recommendations by the National Council of Animal Experimentation Control (*Conselho Nacional de Controle da Experimentação Animal*-CONCEA) for the protection of animals used in experimentation and other scientific purposes.

The trial was carried out in an experimental area located at the Study Group on Forage Farming and Ruminant Production (GEFORP) at the Academic Unit Specialized in Agrarian Sciences of UFRN, in the city of Macaíba, RN, Brazil, and lasted for 94 days, split into 14 days for adaptation and 80 days of data collection. The total area of experimental *Panicum maximum* cv. Massai pasture was 0.96 ha (9600 m^2^) and was equipped with drinking troughs. Concentrate supplementation was provided individually in a management barn with 24 bays, all equipped with individual feed and drinking troughs.

Twenty-four Santa Inês sheep were used, being 12 males and 12 females, with mean initial weight and age of 21.0 ± 2.5 kg and 150 ± 30 days, which were identified, vaccinated, and dewormed every 28 days for parasite control. The animals were assigned in a 2 × 2 completely randomized factorial experimental design featuring two sexes and two supplementation frequencies (every day or every other day) for a total of four treatments and six repetitions.

The animals remained on pasture from 8 a.m. to 4 p.m. and were then herded back to the individual bays, where they were provided supplement according to the strategy adopted: daily supplementation (0.7% of live weight) or supplementation every other day (1.4% of live weight). The concentrate provided during the experimental period comprised ground corn kernels (70%), soybean meal (25%), mineral lick (2.5%), urea (1.8%), ammonium sulfate (0.2%), and table salt (0.5%) and was formulated to meet the maintenance requirements and enable mean daily gains of 100 g, according to the NRC [9]. The pasture were managed under the continuous stocking method, with variable stocking rate (there was a change in the weight of the animals in grazing) and pasture sampling cycles of 28 days, in which the samples were obtained by cutting the forage close to the soil in representative areas using a 1 m^2^ metallic square. The chemical composition of the ingredients is shown in Table 1.

Samples of the forage and of the ingredients of the concentrate supplement were pre-dried in a forced air oven and then ground in a grinding mill with 1 mm sieves for analyses of contents of dry matter (DM; method 934.01), mineral matter (MM; method 930.05), crude protein (CP; method 968.06), ether extract (EE; method 920.39), and organic matter (OM = 100 − MM) following the methodology described by the AOAC [10]. Neutral detergent fiber (NDF) was determined according to the methodology described by Detmann et al. [11]. Total carbohydrates (CHOT) were calculated using the equation proposed by Sniffen et al. [12] and non-fiber carbohydrates (NFC) were calculated according to Mertens [13].

To follow the evolution of body weight, adjustment of the concentrate supply, and calculation of the mean daily and total weight gains, the animals were weighed at the beginning of the experimental period and every 14 days until the end of the experiment, always in the morning before they were sent to pasture. Total weight gain (TWG) was obtained from the difference between the final body weight (FW) and the initial body weight. The mean daily weight gain (MDG) was estimated from the relation between the TWG and the number of days of the experimental period, except for the adaptation period.

Voluntary dry matter intake was estimated using the combination of an external indicator (titanium dioxide) and an internal one (NDFi). Fecal dry matter production was estimated by the external indicator, provided orally (capsules) at 2.5 g/day, split into two 1.25 g doses at 8 a.m. and 4 p.m. for 14 days (eight days of adaptation and six of collection). The feces were collected directly from the rectal ampulla once per day at different times over the six days of collection, with a compound sample formed for each animal at the end of the collection period. Titanium dioxide concentrations were determined according to the methodology described by Myers [14].

To assess intake behavior, the animals were submitted to 48 h periods of observation. Scan sampling was used to asses, every ten minutes, the continuous activities of rumination, idling, and bite rate according to the methodology described by Silva et al. [15]. For bite rate, the time (in minutes) the animal took to take 20 bites was recorded and the data were then converted into bites/minute, as described by Hodgson [16]. Afterwards, the feeding efficiency as a function of dry matter (FEE; g/DM/h), rumination efficiency (RUE; gDM/h), feeding time (FT), rumination time (RUMT), and total mastication time (TMT) were calculated according to Burguer et al. [17] using the following equations: FEE = DMI/FT (gDM/h); RUE = DMI/RUMT (gDM/h); TMT = FT + RUMT (min)where DMI = dry matter intake.

The indigestible neutral detergent fiber fraction (NDFi) of the diets and feces was determined by in situ incubation of samples of the ingredients, Masai grass, and feces in the rumen of one sheep for 288 h, following the methodology described by Valente et al. [18]. Neutral detergent fiber content determination followed the methodology described by Detmann et al. [11]. Afterwards, the voluntary dry matter consumption was estimated.

By the end of the experiment, the males were submitted to water fasting for 16 h and then slaughtered. For the slaughter, the animals were stunned via brain concussion and, after unconsciousness was verified, they were bled, skinned, eviscerated, and had the head and feet removed. A potentiometer equipped with an insertion probe (Testo^®^, model 205) was used to measure carcass pH and temperature right after slaughter and 24 h *post mortem* at the *semimembranosus* muscle. The carcasses were weighed to obtain the hot carcass weight (HCW) and were then placed in a cold chamber (4 ± 1 °C) hanging from the common calcaneal tendon for 24 h.

The gastrointestinal tract was weighed full and empty to determine empty body weight (EBW) and biological or true yield [TY(%) = HCW/EBW × 100]. After refrigeration, a tape measure was used to measure the internal and leg lengths, in addition to croup width, which were used to calculate the compactness index of the leg (LCI, quotient between croup width and leg length) and of the carcass (CCI, quotient between cold carcass weight and internal carcass length). After the measurements, the carcasses were weighed again to obtain the cold carcasses weight (CCW) and to calculate the hot carcass yield [HCY(%) = HCW/BWS × 100] and commercial yield [CY(%) = CCW/BWS × 100], where BWS is body weight at slaughter.

The carcasses were sectioned longitudinally and the half-carcasses were weighed. The left-side half-carcasses were split into six anatomic regions, yielding six commercial meat cuts (neck, shoulder, leg, loin, ribs, and breast), whose weights were recorded for calculations of their respective yields [19]. The left-side loin of each animal was stored in vacuum-sealed high-density polyethylene bags and frozen at −18 °C for later laboratory analyses.

The cooking losses and shear force analyses were carried out on the *Longissimus lumborum* muscle (LLM) according to the methodology proposed by Wheeler et al. [20]. The surface color of the thawed LLM samples was measured using a colorimeter (KONICA MINOLTA, CR 400) after 50 min of exposure to atmospheric air at 17 °C. Color was measured three times in the same sample using the CIE L*, a*, b* color space, where L* = luminosity (100 white, 0 black), a* = red content (+ red, − green), and b* = yellow (+ yellow, − blue), C illuminant, observer at 2 degrees, and 8 mm aperture. The averages for L*, a*, and b* were calculated after the three measurements.

Cost analysis employed the methodology described by Holanda Júnior et al. [21], where supplement cost/animal (BRL) = daily intake/animal * supplement cost/kg; labor cost (BRL) = time needed for the distribution of all supplements (considering 1.5 h of labor) * value of labor hour (BRL4.16) * experimental period (80 days)/number of animals (12, considering only males); feeding cost (BRL) = supplement cost + labor cost; cost of kg of meat produced = total cost/TWG × true yield/100).

A completely randomized experimental design was used. The data were submitted to analysis of variance and the effects of sex and supplementation frequencies were assessed by Fischer’s test at 5% probability. The statistical analyses were performed based on the following statistical model: Yijk = μ + Fi + Sj + (F*S) ijβ (Xijk − X) + eijk
where Yijk = observed value of the dependent variable; μ = global average; Fi = effect of supplementation frequency (daily or every other day); Sj = effect of sex (male of female); (F*S)ij = effect of the interaction (supplementation frequency x sex); β(Xijk − X) = effect of the co-variable (initial body weight); eijk = experimental error.

Carcass data were assessed only for the influence of supplementation frequency since the females were not slaughtered.

## 3. Results and Discussion

No effect of the interaction between supplementation frequency and sex was seen over the variables assessed in this study (*p* > 0.05).

An isolated effect of sex was seen over the intake of pasture, dry matter, and nutrients (*p* < 0.05), in which males exhibited higher values than females (Table 2).

The lack of effect observed for concentrate intake is due to the similarity of the amount provided since it was a function of live weight, which was homogeneous in the experimental batch. Since pasture and supplement made up the total diet of the animals and considering the uniformity in pasture intake, the lack of effect on nutrient intake found for animals supplemented daily or every other day can be explained. According to Barbero et al. [2], pasture intake is influenced by supplementation when the doses are high, which would explain the lack of effect on the parameters discussed herein, given the mean daily dose of 0.7% of body weight. Carvalho et al. [22], when working with sheep of a similar genetic group reared on pasture and receiving concentrate supplementation, found higher intake values of dry matter and nutrients than those in the present research. However, it is noteworthy that the higher frequency of the girder structure in the cells of Masai grass competes with the food intake capacity due to the greater resistance to microbial attack, causing a lower rate of rumen evacuation, which might explain the low dry matter intake.

Higher intakes of dry matter and nutrients were found in males. According to Hashimoto et al. [23], male sheep on pasture may have a lower amount of intermuscular fat due to the greater metabolic weight, which may require higher amounts of food to meet the nutritional requirements [9] and, consequently, increase nutrient intake. Nonetheless, the increase in nutrient intake observed in males was not enough to reflect on performance, so that males and females performed similarly, as did the animals supplemented daily or intermittently (*p* > 0.05; Table 3). When investigating different supplementation frequencies for ruminants reared on pasture, Moraes et al. [6] and Canesin et al. [8] found no effect on animal performance, which reiterates the capacity of ruminants to change food intake patterns as a mechanism to adapt to the intermittent supply of substrates to the rumen [24], maintaining constancy of the ruminal environment and of fermentative patterns, thus resulting in similar performances.

Mean daily gain values and, consequently, total gains were higher than those found by Carvalho et al. [25] when assessing the effect of four types of supplementation in sheep reared on *Urochloa brizantha* cv. Marandu pasture. Those authors reported mean gain of 40 g/day, reaching final weight of 26.4 kg in 84 days of the experiment. It is valid and opportune to point out that, even though no effects of supplementation frequency or sex were found on animal performance, reaching 27.5 kg of body weight at 8 months of age is a satisfactory result for sheep meat production systems on pasture, irrespective of sex. That age sees a balance in tissue deposition in the empty body, with a peak in muscle development and moderate deposition of fatty tissue [26], which results in the production of quality meat, as discussed ahead. For females, which are usually incorporated into the breeding herd and used for replacement, reaching 60% of maturity weight at 8 months of age directly entails an age reduction at first delivery since they will be chronologically and physiologically able to mate, according to Simplício and Maia [27]. Hence, the results obtained in this study may be sufficient to contribute to the efficiency of sheep meat production systems on pasture via a supplementation strategy as it enables the slaughter of males at ideal age and weight and allow females to become breeders at a lower age.

Supplementation frequency and sex did not impact (*p* > 0.05) intake behavior parameters or feeding and rumination efficiencies (Table 4). In spite of the higher intakes of dry matter and nutrients observed for males in relation to females, the lack of variation in intake behavior supports the inferences made by Sampaio et al. [28], who concluded that there are few correlations between nutrient intake and intake behavior of ruminants reared on pasture.

Moreira et al. [29] verified factors that impact sheep intake behavior of tropical grasses and praised the effects of the feed, the environment, and the animal itself on behavioral parameters; however, they pointed out that animals on pasture have a considerable ability to change behavior in response of environmental changes. Therefore, the lack of effect of supplementation intermittence can be understood as the result of the adaptative capacity of grazing sheep, which had already been argued by Mezzalira et al. [24] and previously mentioned in this paper.

Carcass characteristics (Table 5) and the weight and yield of meat cuts (Table 6) were not impacted (*p* > 0.05) by supplementation intermittence. Such results are direct reflexes of nutrient intake, which was also not impacted by the supplementation strategy, and translate into the primordial factor of interference on performance: the greater the intake of nutrients, the better performance tends to be, whether as weight gain or as carcass characteristics or the weights of carcasses and cuts, since intake is the natural pathway for acquiring nutrients that will be converted into tissues deposited onto the empty body. Sun et al. [30] also found no alteration in carcass characteristics or sheep meat cuts in the absence of variations in the intake of dry matter and nutrients, whereas Abebe and Tamir [31] reported different carcass characteristics for animals with different intakes of dry matter and nutrients, results that corroborate the ones in this research. 

Cold and hot carcass yields are considerably below the values reported in the literature for Santa Inês sheep. Grandis et al. [32] and Lira et al. [33] worked with sheep of the same breed and found mean hot and cold carcass yields of 46% and 45%, respectively. The low carcass yields confirm and recall the difficult ruminal disappearance rate of Masai grass, as mentioned by Pedreira et al. [3] and already discussed in this paper, which influences the low effect of the 16 h pre-slaughter fast and, consequently, the amount of abiotic content registered as body weight at slaughter. Proof of that is the similarity of the biological or true yield obtained in this work and in the study by Grandis et al. [32] of 52%.

The weights and yields of commercial cuts (Table 6) and the leg and carcass compacity indices are similar to the revised values [34,35] and compliment the potential of the Santa Inês breed for meat production, which is confirmed by the fact that over 60% of the carcass is made up of noble cuts (leg, loin, and shoulder). More than producing meat, it is noteworthy that sheep of that breed are able to remain productive even under less favorable environmental conditions for production. That corroborates Gurgel et al. [36], who recommended the breed for less intensive production systems due to its adaptative capacity in environments with high temperature and extended drought periods. 

Supplementation frequency lowered the luminosity value of the meat (*p* < 0.05) without affecting hue or saturation (a* and b*) levels, cooking losses, or shear force (Table 7). In spite of the variation observed for luminosity, the values obtained for color match those by Leão et al. [37]. That characteristic is important given the influence of luminosity on the visual value consumers attribute to the meat.

The meat qualitative parameters are very similar to those obtained by Nobre et al. [38] when assessing sheep of the same breed and are very likely due to the successful conversion process of muscle into meat, well characterized by the mean final pH reached (5.5). pH variations of sheep meat beyond the range of 5.5 to 5.8 and changes in acidification rate result in significant alterations in meat quality, impacting water-holding capacity, tenderness, technological yield of the meat, as well as resistance to microbial attack, which highlight the importance of care in pre-slaughter management [32].

Specifically regarding shear force, a parameter that has a greater impact on gaining consumer fidelity, the values obtained classify the mead as having medium tenderness, according to Cezar and Sousa [19], and are similar to the results by Castro et al. [39] for sheep of the same breed. The values obtained are obviously higher than those found for the meat of sheep finished under confinement as a result of the greater muscle activity and lower intermuscular fat deposition of animals finished on pasture. Nevertheless, it is important to keep in mind that the most limiting factor for meat tenderness is animal age, which is positively correlated with the collagen deposition rate and negatively with its solubility, thus reducing the tenderness of animals slaughtered late [40]. Therefore, it is inferred that the sheep finished in that production system were finished and slaughtered at an adequate age to produce meat with the quality demanded by the current consumer market.

Lowering production costs, according to Moraes et al. [6], depends not only on lower feed costs, but also on simpler structures related to the transport and distribution of supplements. The results obtained in the cost analysis are presented in Table 8.

In the present work, the analysis revealed that the kilogram of sheep meat can be produced with 15% cost savings if the intermittent supplementation strategy is applied to sheep reared on Masai grass pasture. The economic impact of the strategy is due to the reduction in labor expenses for feeding, which directly reflects on the total cost with feeding.

## 4. Conclusions

Intermittent supplementation (every other day) of Santa Inês sheep managed with Masai grass pasture is recommended since the strategy does not impact nutrient intake, performance, intake behavior, carcass characteristics, or meat quality, but positively impacts the economic viability of the system as it lowers the costs of producing 1 kg of meat by 15%.

## Figures and Tables

**Table 1 animals-13-01267-t001:** Chemical composition of soybean meal, corn, and Masai grass on days 0, 28, 56, and 80 of the experimental period.

Source of Variation	DM ^1^ (%)	MM ^2^ (%)	CP ^3^ (%)	EE ^4^ (%)	NDF ^5^ (%)	ADF ^6^ (%)	LIG ^7^ (%)
Pasture on day 0	56.10	7.97	3.30	1.74	73.79	43.96	6.56
Pasture on day 28	42.45	7.73	3.25	1.36	68.13	40.24	6.43
Pasture on day 56	41.54	7.36	3.22	1.26	75.15	46.59	10.88
Pasture on day 80	43.27	7.40	3.22	1.20	74.45	44.44	7.63
Soybean meal	89.70	7.58	50.00	2.01	36.60	11.25	1.01
Corn (kernel)	88.45	1.89	8.60	6.14	12.81	4.61	0.57

^1^ Dry matter; ^2^ mineral matter; ^3^ crude protein; ^4^ ether extract; ^5^ neutral detergent fiber; ^6^ acid detergent fiber; ^7^ lignin.

**Table 2 animals-13-01267-t002:** Intake of concentrate, pasture, dry matter, and nutrients by sheep submitted to intermittent supplementation on Masai grass pastures.

Variable	Supplementation Frequency		Sex		SEM	*p*		
	Daily	Every Other Day	Male	Female		F	S	F vs. S
CONI (kg/day)	0.17 ^a^	0.17 ^a^	0.17 ^A^	0.17 ^A^	0.241	0.597	0.886	0.090
PASI (kg/day)	0.35 ^a^	0.30 ^a^	0.38 ^A^	0.28 ^B^	1.739	0.202	0.001	0.093
DMI (kg/day)	0.52 ^a^	0.47 ^a^	0.54 ^A^	0.44 ^B^	1.672	0.214	0.008	0.134
DMI (%LW)	2.13 ^a^	1.84 ^a^	2.18 ^A^	1.80 ^B^	7.493	0.071	0.020	0.056
CPI (kg/day)	0.07 ^a^	0.07 ^a^	0.084 ^A^	0.062 ^B^	0.381	0.210	0.011	0.103
NDFI (kg/day)	0.26 ^a^	0.24 ^a^	0.28 ^A^	0.22 ^B^	1.202	0.299	0.018	0.177
NFCI (kg/day)	0.15 ^a^	0.14 ^a^	0.15 ^A^	0.14 ^B^	0.334	0.371	0.016	0.421

CONI = concentrate intake; PASC = pasture intake; DMC = dry matter intake; LW = live weight; OMI = organic matter intake; CPC = crude protein intake; NDFC = neutral detergent fiber intake; NFCC = non-fiber carbohydrate intake; SEM = standard error of the means; F = frequency; S = sex. ^a^ Means followed by different small letters on the same row statistically differ for supplementation frequency; ^A,B^ means followed by different capital letters on the same row statistically differ for sex.

**Table 3 animals-13-01267-t003:** Performance of sheep submitted to intermittent supplementation on Masai grass pasture.

Variable	Supplementation Frequency		Sex		SEM	*p*		
	Daily	Every Other Day	Male	Female		F	S	F vs. S
FW (kg)	27.92	27.23	27.71	27.44	27.750	0.23	0.63	0.89
MDG (kg/day)	0.07	0.06	0.07	0.06	0.338	0.22	0.62	0.89
TWG (kg)	5.67	4.98	5.46	5.19	27.739	0.23	0.63	0.89
FC	8.04	8.18	9.10	7.12	67.228	0.92	0.16	0.28

FW = final weight; MDG = mean daily gain; TWG = total weight gain; FC = feed conversion; SEM = standard error of the mean; F = frequency; S = sex.

**Table 4 animals-13-01267-t004:** Intake behavior and feeding efficiency of sheep submitted to intermittent supplementation on Masai grass pastures.

Variable	Supplementation Frequency		Sex		SEM	*p*		
	Daily	Every Other Day	Male	Female		F	S	F vs. S
FT (min)	832.24	809.42	812.97	828.70	740.581	0.14	0.29	0.10
TMT (min)	1700.39	1640.45	1679.24	1661.59	2114.027	0.17	0.67	0.59
BR (bite/min)	31.55	30.18	31.68	30.05	88.771	0.31	0.25	0.44
RUMT (min)	868.15	831.02	866.28	832.89	1954.453	0.35	0.39	0.23
IDLET (min)	1168.47	1226.53	1186.97	1208.03	2261.058	0.21	0.64	0.70
FEE (gDM/h)	75.04	75.10	80.66	68.48	304.431	1.00	0.08	0.07
RUE (gDM/h)	71.50	72.78	75.31	68.97	289.945	0.83	0.28	0.19

FT = feeding time; TMT = total mastication time; BR = bite rate; RUMT = rumination time; IDLET = idle time; FEE = feeding efficiency; RUE = rumination efficiency; SEM = standard error of the mean; F = frequency; S = sex.

**Table 5 animals-13-01267-t005:** Carcass characteristics and yields of sheep submitted to intermittent supplementation on Masai grass pasture.

Variable	Supplementation Frequency		SEM	*p*
	Daily	Every Other Day		
HCW (kg)	10.58	10.42	35.828	0.82
CCW (kg)	10.11	10.11	36.102	0.99
HCY (%)	37.77	38.64	67.496	0.53
CY (%)	35.88	37.51	78.282	0.32
TY (%)	52.66	52.55	72.132	0.93
LCI	0.60	0.60	0.802	0.84
CCI (kg/cm)	0.17	0.17	0.668	0.92
pH 0 h	6.35	6.45	11.806	0.69
pH 24 h	5.53	5.48	2.257	0.27
Temperature 0 h (°C)	38.10	37.36	30.497	0.25
Temperature 24 h (°C)	5.77	5.74	2.043	0.53

HCW = hot carcass weight; CCW = cold carcass weight; HCY = hot carcass yield; CY = commercial yield; TY = true yield; LCI = Leg compacity index; CCI = carcass compacity index; SEM = standard error of the mean.

**Table 6 animals-13-01267-t006:** Weights and yields of meat cuts of sheep submitted to intermittent supplementation on Masai grass pasture.

Variable	Supplementation Frequency		SEM	*p*
	Daily	Every Other Day		
Neck (kg)	0.545	0.532	2.431	0.78
Shoulder (kg)	0.924	0.960	2.665	0.51
Ribs (kg)	0.822	0.849	3.659	0.71
Loin (kg)	0.475	0.482	2.613	0.90
Leg (kg)	1.738	1.749	7.157	0.94
Breast (kg)	0.632	0.609	2.406	0.65
----Yield (%)----				
Neck	10.62	10.26	16.154	0.29
Shoulder	18.39	18.60	28.030	0.71
Ribs	15.86	16.30	23.860	0.37
Loin	8.93	9.31	26.038	0.48
Leg	33.79	33.76	17.355	0.93
Breast	12.41	11.76	18.385	0.11

SEM = standard error of the mean.

**Table 7 animals-13-01267-t007:** Meat quality of sheep submitted to intermittent supplementation on Masai grass pasture.

Variable	Supplementation Frequency		SEM	*p*
	Daily	Every Other Day		
L*	42.80 ^a^	37.41 ^b^	63.560	0.002
a*	14.45 ^a^	14.15 ^a^	36.450	0.69
b*	8.02 ^a^	6.17 ^a^	46.902	0.08
Cooking losses (%)	43.94 ^a^	47.09 ^a^	144.661	0.30
Shear force (kgf)	2.56 ^a^	3.59 ^a^	25.237	0.07

L* = luminosity; a* = red content; b* = yellow content; SEM = standard error of the mean. ^a,b^ Means followed by different letters are statistically different (*p* < 0.05).

**Table 8 animals-13-01267-t008:** Production costs of sheep submitted to intermittent supplementation on Masai grass pasture for 80 days.

Costs	Supplementation Frequency	
	Daily	Every Other Day
Supplement cost	BRL19.72	BRL19.72
Labor cost for feeding	BRL20.80	BRL10.40
Total feeding cost	BRL40.52	BRL30.12
Cost of meat kg produced	BRL13.57	BRL11.51

## Data Availability

Not applicable.

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
