# Peer review of "Nutrient Intake, Performance, Carcass Characteristics, Meat Quality, and Cost Analysis of Sheep Submitted to Intermittent Supplementation on Masai Grass Pastures"

_animals, 2023, doi:10.3390/ani13071267_

Round 1
Reviewer 1 Report
The article evaluated intermittent supplementation in pasture-finished sheep. The article is very relevant, but needs improvement in the introduction and discussion. Please update the references used in the introduction and discussion.
Author Response
Dear reviewer,
Thank you for reviewing the material.
We sought to use the most current references on the subject, however, we could not despise the classic authors of the theme. Feeding ruminants with grass and supplements involves modern concepts as well as traditional concepts in the study of forage, so we do not hesitate to cite older literature along with the most current ones. This condition is complicated when we talk about feeding meat-producing sheep managed in tropical pastures, because these animals have always been managed in two conditions: free in the pasture or confined. Therefore, finding very new references on the intensive production of sheep meat raised on pasture is difficult, because the intensive production of sheep on pasture is new in tropical regions. The main articles on the subject are cited in our manuscript.
Best regards,
Stela Antas Urbano.
Reviewer 2 Report
Suggestions in attachment.

Author Response
Dear Reviewer,
We are grateful for the careful review of our manuscript and for the desire to contribute to the improvement of the material. We inform you that all requests for corrections have been accepted and changes have been made to the manuscript that is attached.
- "Consumo"' has been replaced by "intake" throughout the manuscript;
- We increased the description of the grazing method to try to make it more didactic;
- We made it clear in the text that ONE fistulated animal was used in the digestibility test;
- We "cleaned" table 2 to make it more objective. Thanks for the note, it looks much better after the suggested adjustment;
- We describe all carcass indicators in the methodology. LCI and CCI were missing the abbreviations in the text.
We understand that the concentrate intake is not a result, but we chose to leave it there because the DMI was the result of pasture intake + concentrate intake, so the different numbers of DMI and PASI could lead to confusion in the understanding if the CONI values do not were presented. However, we reiterate that we are available to eliminate the CONI from the Table if the review committee understands that it is harmful to the manuscript.
Thanks again for the good review work!
Best Regards,
Stela Antas Urbano
Reviewer 3 Report
The manuscript entitled "Nutrient Consumption, Performance, Carcass Characteristics, Meat Quality, and Cost Analysis of Sheep Submitted to Intermittent Supplementation on Masai Grass Pastures" written by Urbano et al. attempts to investigate the effect of supplemental feeding of concentrate on sheep production.
This manuscript was well written and does have data that is well presented. However, the general objectives of the paper was covered already known and it is known that increasing supplementation with concentrated feed will increase economic efficiency (less time to reach market weight). The only information the article adds will be that supplementation every other day can be employed with this breed of sheep under these conditions as it is more economical.
The introduction was well written with a clear concise background of the work done before. The objectives were also clearly stated and addressed with appropriate methods.
The tables were well constructed and the results were clear and easy to read. The discussion should have been separated from the results (in my opinion).
Author Response
Dear Reviewer,
Thank you for carefully reviewing our material.
With great respect, I would like to clarify that we have not investigated the effect of the amount of supplement offered on production efficiency. The amount of concentrate was the same for all animals. Supported by the stability of the ruminal environment, we tested two supply strategies (daily or on alternate days) in an attempt to reduce production costs arising from labor costs. The interaction between pasture and supplement is dynamic, especially in sheep, which have a specific grazing habit. We focused on this point of investigation because the elucidation of points like these could contribute to the sheep meat production chain.
We provide a review of the material and the proof is attached.
Best Regards,
Stela Antas Urbano

Round 2
Reviewer 1 Report
The authors complied with and/or justified the proposed changes.
The manuscript can be forwarded for publication.